# Evaluation of the Sustainable Development of Macau, Based on the BP Neural Network

**Yue Huang** [1,2] , **Youping Teng** [2] **and Shuai Yang** [1,2,*]

1    College of Computer Science and Technology, Zhejiang University, Hangzhou 310063, China
2    College of Art and Archaeology, Zhejiang University City College, Hangzhou 310015, China
*    Correspondence: samyang@zju.edu.cn; Tel.: +86-158-7566-3395

**Abstract:** (1) Background: the rapid development of cities and the process of industrialization has improved the level of economic development for all humanity, accompanied by a series of problems, such as the waste of ecological resources and the environmental destruction. Macau has long been one of the regions with the most active economic activities. However, the phenomenon of economic recession, unreasonable land use, and frequent flood disasters have appeared in the past few decades, which violate the concept of sustainable development; (2) Methods: this paper uses the BP neural network model to evaluate the sustainable development of Macau; (3) Results: the weight ranking of sustainable development is: economic > social > ecological environment. A correlation analysis shows that ecology and economy have a significant negative correlation; (4) Conclusions: In Macau, economic growth is given priority while social and ecological environment development lags behind. Macau has problems, such as a unitary economic structure and a high population density. This paper advocates that investment in ecological protection should be increased, environmental resources should be optimized and saved, and the concept of sustainable development should be strengthened in the application of Macau's urban development.

**Keywords:** sustainable development; Macau; ecological; economy; weight assessment

## 1. Introduction

A city is a place where human beings have lived for a long time, which has a history of five thousand years [1]. It is a kind of living environment with purpose, planning, and a highly artificial natural environment created by the human lifestyle, for many years [2]. It is destined to be a complex comprehensive system integrating society, natural environment, and economy [3]. Many countries are concerned about sustainable development, and the level at which each country has reached attests to their efforts, and sustainable issues have a direct bearing on the relevant well-being of the country [4]. The sustainable urban development is to integrate the social, environmental, and economical development of a city, to achieve a stable and durable state. The core of sustainability is development [5]. It should not only achieve the goal of economic development, but also promote the continuous progress of the social system and the improvement of the population's quality education. At the same time, it should also protect the natural resources and environment upon which human survival depends [5]. For a city, it is more direct to find a rapid development path with a similar sustainability from various aspects of the economy, society, and environment [6]. Sustainable development is the frontier and a hot spot in the field of development. Due to resource decay and the lack of overall planning, many cities have accumulated contradictions and problems in their development [7]. For example, Macau (a Chinese city), which is highly dependent on the gambling industry, has gradually highlighted the characteristics of economic, social, and ecological environmental vulnerability, and the problem of unsustainable development has become increasingly prominent. In particular, a series of problems, such as the deterioration of the ecological environment,

the unreasonable economic and industrial structures, and the lack of industrial continuity, have greatly threatened and challenged the benign development of the city.

From the perspective of economic development, Macau's economy takes gambling as the industrial pillar, which is called the "Oriental Las Vegas". Gambling and tourism have brought considerable economic income to Macau [8]. Some cities take gambling as an important economic industry. Las Vegas is a typical representative, with theme hotels as the main carrier, and a variety of carriers supporting to promote its common development. In Las Vegas, it is not just gambling that attracts tourists. Leisure is also a major factor. The Las Vegas Strip and Flamingo Road are the main shopping streets, surrounded by most of the theme hotels, shopping centers, bars, and small hotels. The commercial street has formed a certain scale, attracting countless visitors for sightseeing [9]. Las Vegas has transformed itself from a "gambling city" to a "tourist city" [10]. Monte Carlo, east of Monaco, is known for its luxurious casinos and is one of the world's top three gambling cities, along with Las Vegas and Macau [11]. The boom in gambling, along with the development of Monaco tourism, banking, real estate, and other industries. In addition, activities, such as the Monte Carlo International Acrobatic Festival, International Fireworks Festival, and Formula One Grand Prix are famous around the world, as well as a large number of other cultural and sports activities that attract tourists. The Government of Singapore has maintained a strong leadership in the development of the gambling and promoted gambling as one of the crucial industries in Singapore's diversified economy. Marina Bay Sands, an integrated resort developed by Las Vegas Sands, is reputed to be the world's most expensive stand-alone casino building. The Marina Bay Sands complex has transformed Singapore's skyline with a bizarre "hanging garden" on its roof. Furthermore, the government vigorously promotes Singapore's scenic spots overseas, and is developing marine cruise tourism, medical tourism, educational tourism, sports tourism, etc., and is building Singapore into one of the world's top three convention and exhibition industry centers [12]. The booming tourist market of Miami, where a large number of immigrants settled and made the city famous as a "multicultural center", now attracts many tourists from Europe, Canada, Asia, and other places, to experience the exotic environment. Miami has taken advantage of its scarce coastal resources to develop tourism, with casinos and bars and entertainment facilities on cruise ships, and the economic benefits are considerable. Miami also focuses on hosting major events and conferences. Now, Miami has become a world-class tourist city through the development of tourism and the related industries [13].

From the perspective of ecological development, China has developed very rapidly and the urbanization is also accelerating in recent years [14]. In building cities, people often pay too much attention to economic development and neglect the ecological construction of cities. The consequences of deviating from nature have also come one after another [15]. There are concepts, such as the ecological city, compact city, three-dimensional city, resilient city, livable city, green city, landscape city, and low-carbon city, which have been pointed out. Multi-dimensional urban construction has become the mainstream of urban planning [16]. In recent years, through a literature review, it has been found that there are extensive studies on the co-development of urban ecology and economy. It shows that sustainable development is a multi-party process of coordinated development and a circle with a causal effect. The result of the coordination is the synchronous development of ecological, social, and economic benefits, which is a virtuous cycle, and vice versa [17].

Based on the main line of ecology and economy, this paper uses the back propagation (BP) neural network model to evaluate the sustainable development of Macau (China). It analyzes the synergy degree of the sustainable development of Macau. In addition, based on the evaluation score of Macau's official data, it is concluded that there is a significant negative correlation between Macau's ecological and economic development, which verifies the necessity of implementing the sustainable development renewal strategy in Macau. The above research has an enlightening significance for grasping Macau's renewal plan and giving play to the advantages of a sustainable development strategy. It can provide some reference for Macau and even more the regional renewal.

## 2. Literature Review

### 2.1. Sustainable Urban Development

In the late 18th century, with the rapid economic development brought about by the industrial revolution, the damage to the ecological environment was obvious, and the harmony between man and nature was no longer harmonious [18]. As Lewis Mumford noted: between 1820 and 1900, the destruction and chaos in the great cities was almost as great as on the battlefield [19]. The "Earth limit theory" was put forward in 1972, and its method of thinking is similar to the concept of "environmental capacity" in ecology. Environmental capacity refers to the upper limit of the growth of the biological population, and the population stops growing when it reaches the upper limit. D.H. Meadows et al. argue that a similar pattern exists for the socioeconomic development. The so-called limit theory actually refers to the environmental capacity of social and economic development [20]. Later, the Club of Rome made an in-depth study and a refinement of the "catastrophic collapse of mankind", and changed the "zero-growth view" to the "organic growth concept" [21]. In the same year, the United Nations held the World Environment Conference for the first time and pointed out the idea of sustainable development [22]. In 1983, the United Nations established the Commission on Sustainable Development (UNCSD). According to the UNESCO, "Man and Biosphere Program" (MAB), which put forward EI (ecolite infrastructure) as an official concept, and formulated the principles of eco-city planning [23]. In 1987, the UNCSD provided the first authoritative interpretation of the concept of sustainable development in its report "Our Common Future". It was determined that "sustainable development is the development that can, not only meet the needs of the present generation, but also do no harm to meet the needs of future generations" [24]. Sustainable development gradually gained attention in urban ecology, economy, society, planning, and other disciplines [25], and rapidly developed into an important research topic. This definition has been widely recognized, and its core is that the premise of global economic and social development is not to exceed the carrying capacity of the resources and the environment [26,27]. Campbell [28] believes that social equity, economic development, and environmental factors are the three indexes in sustainable urban development planning, and it is an important issue to consider in planning to solve the contradictions between the indicators, respectively. Honachefsky [29] interpreted the concept of ecological orientation, pointing out that the destruction of the ecological environment is the result of economic development, and put forward the idea of "ecological optimization". The concept of a complex ecosystem was first proposed by Ma Shijun and Wang Rusong [30]. They believed that nature, economy, and society constituted this complex ecosystem, and the relationship among the three should be causal, complementary, and balanced. Chinese experts and scholars have also closely followed the pace of global research on sustainable development. Since the mid-to-late 1980s, many researchers have conducted an in-depth exploration of sustainable development, especially in capacity building and evaluation. For example, Haileti et al. [31] established an index system for the comprehensive evaluation of the sustainability, coordination, and development levels of a city. They applied a fuzzy comprehensive evaluation method and analytical hierarchy process to quantitatively evaluate the sustainable development of major cities in China. Based on the previous evaluation system, Liu Song and Liu Binyi [32] established the principle of the human-oriented evaluation index system for sustainable development. A set of evaluation indexes for the sustainable development of an urban human settlement environment is proposed by using an analytical hierarchy process. Xu Xueqiang and Zhang Junjun [33] used a quantitative method to comprehensively evaluate the coordination index of the environmental, economic, and social developments in Guangzhou, over the course of 17 years. It has been found that the sustainable development of Guangzhou is evolving to a higher level, and the changes in the environment, economy, and society are not in sync. Through the case summary, sustainable development is mainly based on three aspects: (1) using the conceptual model for the empirical research, (2) an evaluation, based on the latest evaluation methods and technical capabilities, and (3) a typical regional study of the provincial administrative regions [34].

Le Corbusier [35] proposed the concept of the Ville Radieuse. With the development of science and technology, the establishment of space models and other new technologies have been applied to urban planning and construction. Eero Saarinen [36] interpreted the theory of organic evacuation and a large number of practical applications. I. McHarg [37] proposed the ecological design method of "design with nature" and carried out a lot of practice. The "mother-and-child city" built by Mitsui in Japan was a model of implementing the concept of sustainable development [38]. The theory of the "garden city" was explained by Ebenezer Howard, as early as 1898. He advocated the integration of city and nature, aiming to formulate a set of unified standards for the urban form, which could, not only make the orderly and stable development of the city, but also solve the problems of urban ecology and environment [39]. Timothy Beatley [40] proposed the concept of green urbanism, in which he proposed to apply the concept of resilience to various aspects of urban energy, water resources, transportation, and infrastructure, so that cities can actively cope with adverse factors brought by the natural environment. The International Council for Local Environmental Initiatives (ICLEI) put forward the concept of "resilience" at the UN Global Summit on Sustainable Development. A resilient city refers to a city that has enough capacity to accommodate and sustain the pressure brought by social, economic, environmental, and technological developments, today and in the future, and the infrastructure planning can still play the necessary functions in the future [41]. Concepts, such as the sponge city, proposed in the 2012 Low-carbon City and Regional Development Science and Technology Forum, are good interpretations of the ideal spatial model of sustainable urban development. These theories also explain the concept of sustainable urban development from different perspectives [42].

In this paper, by extracting the keywords from two periodical databases (CNKI and WOS) in the past 15 years, it has been found that the research field on sustainable urban development has continued to expand and has been refined to include the urban ecology, living environment, urban circular economy, economic transformation, and industrial structure, as well as moving in other directions. The concepts of urban culture, creative economy, low-carbon city, smart city, compact city, shrinking city, ecological city, and the green city, came into being one after another, which were the supplement and sublimation of the theory of sustainable urban development. Some researchers believe that these urban concepts are models of sustainable urban forms [43]. Scholars, such as Junli Li proposed that the construction of a circular economy eco-city (CEE) is the most effective way to solve the problem of sustainable urban development [44]. Scholarly research on the sustainable development of cities have begun to pay attention to the quality of urban life, which covers the ecological, economic, political, social, and other aspects. In addition, it discusses how to not increase the burden for future generations in the subsequent development mode, so that the ecological, economic, social, and other fields can realize their sustainable development in the city's development [45]. Urban sustainable development has become an important topic of academic research. On the one hand, the expansion of cities is indicative of the growth of the overall socio-economic development of the world. On the other hand, it raises a number of economic, social, and environmental problems [46]. Xiangzheng Deng et al. [47] discussed the scientific connotation of a balanced regional development, and proposed to solve the imbalance between the economy, humans, and nature, by citing the theory of sustainable development. With the concept of development geography, they examined the road of the balanced regional development in China from the perspectives of society, economy, and ecology. Zongwei Han et al. [48] believe that there are still challenges to addressing the negative impacts of urban expansion on the ecological environment and economic development. Zhiqiang Wu [49] proposed that the social, economic, and ecological interactions of cities are the three most important elements to realize the sustainability. The three elements are not parallel relations, but are the mutual transformation and support relations. For example, the transformation of social factors into economic factors, the transformation of ecological factors into social factors, etc. It is crucial to realize the change, and promote the integration of the various

factors. As these factors are unavoidable problems, many studies define urban sustainability as the analysis of economic, social, and the environmental sustainable development, and pursue the interaction and coordinated development of the three "pillars" [50]. Based on the challenges of sustainable development in modern cities, the New Leipzig Charter, adopted in 2020, describes the innovative approaches to urban development. It shifts the focus from the practical solutions to urban development, to a broader vision. Some scholars have even suggested to the administrative authorities that urban development needs the encouragement and support of the state. Governments should consider how the SDGs can be used to influence day-to-day decision-making and should prioritize the implementation of the SDG-oriented urban infrastructure and plans [51]. Sustainable urban development should be regarded as a comprehensive and complex urban system, and economic development is the driving force of urban development. However, a synchronous economic development should ensure the ecological integrity, strengthen the resistance to natural disasters, and optimize urban resources to ensure the sustainable use. Moreover, how can the protection and utilization of social and cultural resources improve the quality of public life? All of these problems can be included in the research on sustainable urban development [52].

### 2.2. Evaluation of the Sustainable Development

In this paper, the indicator system of the sustainable development is constructed, referring to many works of literature. Once *Agenda 21* was adopted at The United Nations Conference on Environment and Development (UNCED), in 1992, The issue of building an indicator system for sustainable development gradually attracted extensive attention from various countries, organizations, and academic circles [53]. In 2007, the UNCSD published *The Sustainable Development Indicators: Guidelines and Methodology* [54]. This indicator system is summarized from the three levels of economy, society, and environment, and 14 general goal systems of sustainable development are extracted, including 44 sub-goals and 96 indicators. In addition, the Sustainable Development Goals (SDGs) have been improved upon in *The 2030 Agenda for Sustainable Development* of The United Nations [55]. The above definition of the indicators is the primary basis for selecting the indicators in this paper. Although the framework is designed for the national level, international organizations, and research institutions, various levels of government have actively carried out localization work within the framework of the global indicators [44]. In recent years, 193 countries have worked towards achieving the UN's 17 Sustainable Development Goals (SDGs). These targets cover a range of issues, from poverty to gender equality and climate change [56]. Biekša, K. et al. analyzed the sustainable economic development of EU countries, according to the SDGs and used the integrated sustainability and environmental footprint index. They argued that sustainable economies can be driven by economic, environmental, and social dimensions, applying the principles of innovation and knowledge [57]. In 2019, the UNSDSN selected 15 goals and their related indicators, according to the official SDG indicator framework, to evaluate the progress of the SDGs in some key cities in the United States and Europe, and built a basic method system for monitoring and evaluating the SDGs at the city level [58]. Stoenoiu, C.E et al. extracted nine indicators from the SDGs to evaluate the sustainable development of eight Eastern European countries, between 2013 and 2019, to classify them into high, middle, and low progressive countries, and ranked them [4].

With the implementation of the SDGs, scholarly research on sustainability integrates previous studies on the SDGs to further explore the sustainable development, and Chinese scholars also continue to explore and study the localization, according to the SDGs [59]. In the practical application, the global implementation of the SDG framework of "classification, over-all planning, and collaboration", is gradually improved upon [60]. From different perspectives, many people also explained an open index system framework of the urban sustainability evaluation, from five dimensions: urban inclusiveness, urban safety, urban cheapness, urban resilience, and urban cleanliness [61]. Wang Penglong et al. also conducted in-depth research

on the progress of the SDGs' sustainable urban development evaluation index system, they identified the critical fields of the sustainable city construction, and established an open system framework for the urban sustainability evaluation [62].

### 2.3. Sustainable Development of Macao

The research scope of this paper covers the whole territory of Macao. The Macao Special Administrative Region includes Macao Peninsula, Taipa Island and Coloane Island, and the newly reclaimed area (as shown in Figure 1). Macau is adjacent to Zhuhai, China, with a local population of 683,200 and a total land area of 32.9 square kilometers (2020.12 GMT).

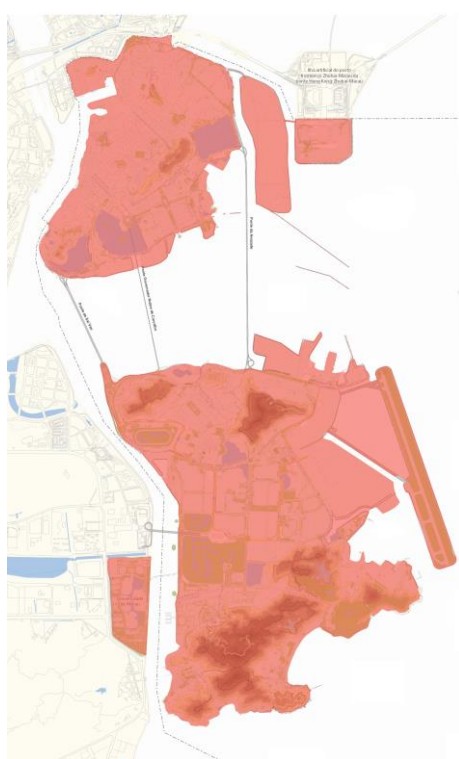

**Figure 1.** Study area. Source: The authors.

Although Macao is a tiny place with a small population and area, it has created a series of economic miracles and has become one of the richest cities in the world [63]. Under the influence of historical and cultural factors and government policies, Macao's gambling industry has gradually become a leading economic force. The gambling industry and Macao's economy have formed a symbiotic pattern of mutual prosperity and loss, which has caused a very large negative impact on the social and environmental aspects [64]. Macau's over-reliance on gambling has long been a problem. Reviewing the sustainable development of Macao's economy, Wu Jiangqiu et al. [65] advocated that the diversification strategy of Macao must focus on the development of industry clusters related to the gambling industry, including the tourism and exhibition industries, so as to realize the coordinated development of the gambling industry and other industries.

Macao is surrounded by the sea on three sides, and there are a lot of flood disasters. In 2017, Typhoon Hato caused serious floods in the coastal areas of Macao, especially in the inner harbor area. Macao's ecological, economic, and social development is unbalanced. Chen Ping et al. [66] analyzed the Macao government's solutions to the vulnerability of the urban system. The author believes that although the government has implemented a series of governance measures, including land reclamation, land expansion from Hengqin, the moderately diversified development of emerging industries, and the introduction of foreign workers under the priority of the local residents' employment, and so on, which have achieved some results, the sustainable development of Macao's urban system is still

insufficient. It is urgent for Macao to carry out urban planning with a larger spatial scale, resources, and new mechanisms. Wu Mingwan et al. [67] analyzed the relationship between the green economy and resource and the environment carrying capacity in the Greater Bay Area. Based on the data of the statistical yearbook, the comprehensive ECC index system of the economy, resources, and environment was constructed, and the coupling relationship between the economy, resources, and environment system in Macao was obtained. Wang Yening et al. [68] evaluated the ecological carrying capacity of the Guangdong-Hong Kong-Macao Greater Bay Area. Among them, Macao is in the lowest position among the nine cities, due to the shortage of natural capital flow and the depletion of the accumulated stock.

Based on the scientific concept of the urban sustainable development, this paper evaluates the sustainable development status of Macao and analyzes the ecological economic and social development status of Macao in multiple dimensions. It provides an important objective basis for the subsequent urban renewal of Macao. By the way, although Macao's development model is unique and unreplicable, in practice, an in-depth interpretation of its sustainable development trajectory is of great strategic significance, to help Macao find its own position in the Guangdong-Hong Kong-Macao Greater Bay Area, the Greater China economic circle and even the global cities.

## 3. Methodology

### 3.1. Research Theory

The 2030 Agenda for Sustainable Development of the United Nations has set 17 Sustainable Development Goals (SDGs) and 169 hierarchical indicators [55]. The implementing of this index system aims to thoroughly solve the development problems of the three dimensions of society, economy, and environment. The SDGs are committed to the inclusive growth and coordinated economic, social, and environmental development. They guide the development policies and the use of funds in countries around the world, up until 2030. They lead humanity to take action in areas that are critical to the planet, to end poverty, protect the world, and ensure prosperity [69]. Based on the index framework of the SDGs, this paper constructs the index system of the sustainable development for Macau [70].

### 3.2. Evaluation System

Economic growth has indeed brought a lot of benefits, such as the improvement of people's quality of life and the improvement of living standards. However, the ecological environment in which people live is also one of the crucial factors of sustainable development. How to develop the economy without damaging the ecology, or even to better maintain the ecological environment and realize the win-win relationship between the ecology and the economy. This is a topic worth discussing.

Based on the theory of sustainable development, this paper takes the synergy between the various elements, as the main research argument and approach. On this basis, the theory of urban ecological economics has been widely recognized and studied by the academic circle, and a relatively perfect theoretical system has been formed. This paper tries to make a comparative analysis of Macau's ecology and economy, to find targeted planning strategies.

This paper uses the BP neural network model to evaluate the sustainable development of Macau. The purpose is to evaluate the weight of the economic, ecological, and social development in Macau's sustainable urban development and to explore the relationship between the ecology and economy in Macau. There are many kinds of evaluations of sustainable development indicators, each with its advantages and disadvantages. Some quantitative methods, such as the analytic hierarchy process (AHP) and the Delphi method, which relied a scoring system. Therefore, there are different degrees of subjective judgment, which may affect the scientific nature of the results. The ecological footprint method is scientific and operable, but it is not easy to reflect the sustainable state of the system

completely. In this paper, the relevant institutions of the Macau government provided data for the sustainable development indicators of Macau, which provided an objective basis for the quantitative calculation and correlation analysis (the institutions are discussed in Section 4.1). The BP neural network model does not need to incorporate any subjective factors to ensure the objectivity and accuracy of the quantitative analysis of Macau's sustainable urban development. Furthermore, a Pearson correlation analysis on Macau's economic 441 indicators and the ecological environment indicators is undertaken.

The artificial neural network (ANN) is a kind of model designed according to the structure characteristics of the biological neural networks. There are many artificial neural network models which have been developed and that are widely used at present, such as the BP neural network, the Hamming network, the Grossberg network, the Elman network, and the Hopfield network [71]. The BP neural network is one of the most commonly used models in the artificial neural network model. To use this model in the construction of various evaluation systems, it is necessary to use the fitting neural network model to obtain the weights and bias values of each layer in the network, in order to determine the weight of each indicator in the evaluation system [72]. Compared with the traditional statistical model, this kind of network has a stronger fault tolerance, self-adaptability, and self-organization.

The BP neural network is also called the error backpropagation algorithm. It is a multi-layer feed forward neural network, where, the input data is transmitted forward, and the fitting error is transmitted backward. In the process of the forward transmission, the input data passes through the input layer, the hidden layer, and finally reaches the output layer (Figure 2). The hidden layer of the BP neural network can have one layer or multiple layers. There are different numbers of neurons in each hidden layer. If a layer of neurons cannot obtain a reasonable output after the calculation, the layer of neurons will back-propagate the calculated error. Thus, the weights and thresholds between the neurons in this layer and the input data are adjusted to make the actual output of the BP neural network close to the desired output [73]. The BP neural network has the ability to arbitrarily make complex pattern classifications and map excellent multi-dimensional functions. It can solve the exclusive OR (xor, a mathematical operator) and some other problems that a simple perceptron cannot solve. It is mature in both network theory and performance, and its outstanding advantages lie in its strong nonlinear mapping ability and flexible network structure [74]. The following takes the simplest two-layer neural network with one hidden layer and one output layer, as an example, the structure of the BP neural network is introduced.

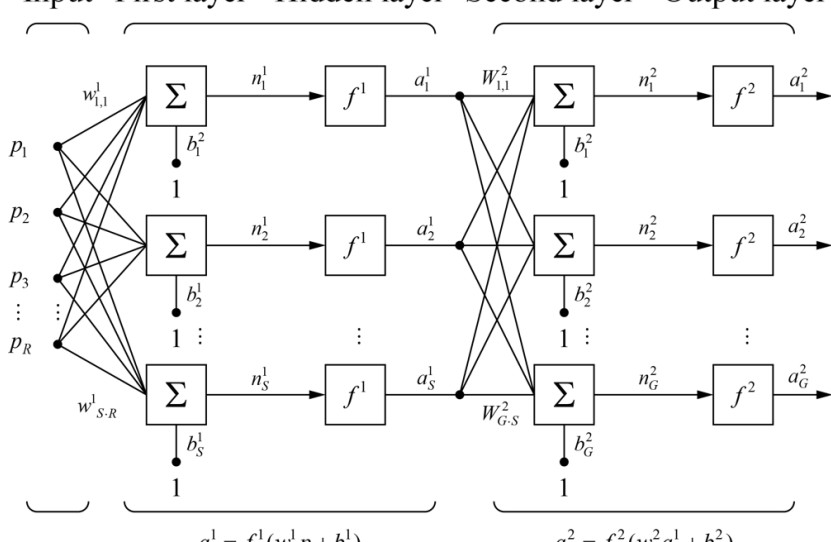

**Figure 2.** Structure of the BP neural network. Source: The authors.

As shown in Figure 2, in the two-layer neural networks, the input of the first layer is the input of the whole neural network system. Assuming that there are R input indicators, the input indicator is $\{p_1, p_2, \ldots, p_{R-1}, p_R\}$. The hidden layer of the first layer assumes a total of S neurons, denoted by the symbol "Σ". There is also a bias value (threshold) below each neuron, denoted by the symbol $\{b_1, b_2, \ldots, b_{S-1}, b_S\}$. The output of each neuron in the first layer is a function, representing the weight of each indicator, the product sum of the input, and the threshold value [75]. For example, the output calculation formula of the first neuron in the first layer is:

$$n_1^1 = \sum_{i=1}^{R} w_{1i}^1 p_i + b_1^1$$

The output calculation formula of the remaining neurons in the first layer can be shown, as follows:

$$n_j^1 = \sum_{i=1}^{R} w_{ji}^1 p_i + b_j^1, \ (j = 1, 2, 3, \ldots, S)$$

There is a transfer function F between the output of each neuron in the first layer and the total output of this layer. The transfer function is also called the start function. The total output of the first layer is the input of the second layer. The calculation formula of the total output of the first layer can be shown, as follows:

$$a_j^1 = f^1\left(n_j^1\right), \ (j = 1, 2, 3, \ldots, S)$$

The structure of the second layer is the same as that of the first layer. The input of the second layer first obtains the output of each neuron in the second layer through the product sum formula, and then the output of each neuron gets the total output of the second layer through the transfer function. Suppose there are $G$ neurons in the second layer, then, the calculation process is, as follows:

$$n_k^2 = \sum_{j=1}^{S} w_{kj}^2 a_j^1 + b_k^2, \ (k = 1, 2, 3, \ldots, G)$$

$$a_k^2 = f^2\left(n_k^2\right), \ (k = 1, 2, 3, \ldots, G)$$

Through the above calculation formula, the total output of the second layer can be obtained. Since the neural network in this example has two layers, the total data in the second layer is the output of the entire neural network.

### 3.3. Transfer Function Type

There are two common transfer functions in the BP neural network: the S-type transfer function and the linear transfer function. The S-type transfer function is generally used in the Sigmoid function, and the Sigmoid function can be divided into the *log-sigmoid* function and the *tan-sigmoid* function [76]. The two functions are, as follows:

$$Log - Sigmoid(x) = \frac{1}{1 + exp(-x)}$$

$$Tan - Sigmoid(x) = \frac{2}{(1 + exp\left(-2 \cdots x\right))} - 1$$

The difference between them is that the log sigmoid function's prediction result is in the (0,1) interval, while the tan sigmoid function's prediction result is in the (−1,1) interval. The curves of the two transfer functions are shown in Figure 3.

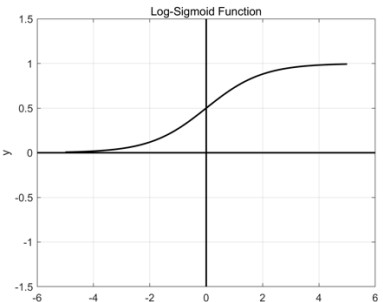
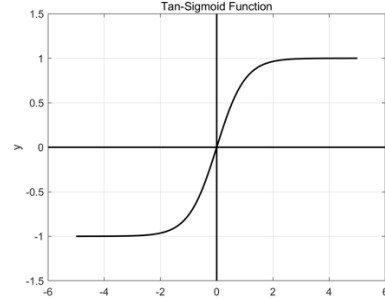

**Figure 3.** Linear transfer function diagram (**left**: log sigmoid function prediction result interval (0,1) **right**: tan sigmoid function prediction result interval (−1,1)).

## 4. Results

### 4.1. The Construction of the Sustainable Development Indicator System

The construction of the sustainable development indicator system is mainly aimed at Macau, to apply the BP neural network model to the evaluation system of the sustainable development of Macau.

The selection of the indicators in this paper is based on two aspects. Firstly, the UNCSD Sustainable Development indicator system and the 17 Sustainable Development Goals (SDGs) of the United Nations were used as the reference systems and is based on the regional characteristics of Macau's sustainable development, according to the official data of Macau from 2011 to 2018. Specifically, it includes the *Report on the State of Macau's Environment*, The Statistical Yearbook and Macau Environmental Statistics, and *Macau Industrial Structure Report* published by the Macau Statistics and Census Service (DSEC) (http://www.dsec.gov.mo, accessed on 1 October 2021). These are analysis reports on the statistical indicator system of the moderate and diversified development of the Macau economy, etc. [77,78].

Secondly, some literature has expounded on the sustainable development indicator system of Macau. The indicator extraction in the literature refers to the sustainable development indicator system of some European countries, such as France and the United Kingdom, as well as the United States. This paper also draws on the indicator system of the sustainable development in Taiwan (China). It focuses on *The Evaluation of the Sustainable Development of Hong Kong, Macau, and the Pearl River Delta* proposed by the research group at Sun Yat-sen University in Guangzhou [79]. On this basis, a set of indicators in line with the actual situation in Macau was created (Table 1) to evaluate the sustainable development of Macau [80]. The proposal of this indicator system can comprehensively assess and monitor the sustainable development of Macau.

Based on the above two points, this paper draws on the indicators of Macau's sustainable development, and adjusts the selection of the individual indicators, according to the key points of sustainable development. For example, in the indicator system of the economic sustainability, Macau's gambling industry is an essential economic development indicator, so the proportion of the gambling industry in the GDP is increased. In addition, for the currently unpublished indicator data, they can be replaced with the similar or the same data reflecting this indicator. For example, the "air compliance rate" indicator is not published in the official data of Macau, so it is considered to replace it with a similar "good air rate" indicator. It can reflect the air quality level of Macau. Detailed indicators are shown in Table 2.

**Table 1.** Feasible structure of the Macau sustainable development indicator system.

| Rule of Indicators | Indicator Field | Indicator Layer |
|---|---|---|
| Sustainable development level | Level of economic development | |
| | Economies of scale | GDP |
| | | Annual GDP growth rate |
| | Economic benefits | GDP per capita |
| | | GDP per square kilometer |
| | Economic structure | Secondary industry coefficient |
| | | Tertiary industry coefficient |
| | Economic outgoing | Per capita flow of foreign direct investment |
| | | Import/export as a share of GDP |
| | Level of social development | |
| | Population indicator | Natural population growth rate |
| | | The population density |
| | Quality of life | Number of hospital beds per 10,000 people |
| | | Books per 10,000 people |
| | | Median job income |
| | Urban infrastructure | Road length per capita |
| | | Number of telephone subscribers per 10,000 people |
| | | Cargo volume per capita |
| | Social stability and security level | Local unemployment rate |
| | | Crime rate |
| | Resource and environmental support levels | |
| | Resource conditions | Per capita electricity consumption |
| | | Per capita water consumption |
| | Ecological environment | Solid waste emissions per capita |
| | | Wastewater discharge per capita |
| | | Per capita area of public green space |
| | Resource and environmental pressures | Resource |
| | | Environmental stress |

Source: The authors.

This paper constructed an evaluation indicator system of Macau's sustainable development level, including the social, economic, and ecological dimensions. The social dimension includes nine evaluation indicators, the economic dimension includes eight evaluation indicators, and the ecological environment dimension includes eight evaluation indicators. This paper collected data from 25 measurement indicators in Macau from 2011 to 2018. The original data of 25 indicators are shown in Table 3.

*4.2. Results and Analysis of the Artificial Neural Network*
4.2.1. Standardized Processing of the Original Data

The evaluation indicator selected in this study has a wide range of sources, and the data units are different. To avoid the distortion of the model fitting results caused by the significant unit differences, this study uses the range standardization method to process the original data. The formula is, as follows:

$$stdx_{ij}(+) = \frac{x_{ij} - min_i}{max_i - min_i}$$

$$stdx_{ij}(-) = \frac{max_i - x_{ij}}{max_i - min_i}$$

**Table 2.** Evaluation indicator system of the sustainable urban development in Macau.

| The Target Layer | Rule Layer | Indicator Layer | Unit |
|---|---|---|---|
| A—Macau is acceptable Level of sustainable development | B1—social | C1—Natural growth rate of the population<br>C2—Urban population density<br>C3—Number of hospital beds<br>C4—Books<br>C5—Median monthly job income<br>C6—Total length of road lanes<br>C7—Per capita living area<br>C8—Unemployment<br>C9—Crime growth rate | One over one thousand<br>People per square kilometer<br>Pieces<br>Number<br>Patacas<br>km<br>square meters<br>%<br>% |
| | B2—economic | C10—Gross regional product (GDP)<br>C11—GDP growth<br>C12—GDP per capita<br>C13—Second industrialization coefficient<br>C14—Third industrialization coefficient<br>C15—Gaming as a share of the GDP<br>C16—Foreign direct investment<br>C17—Ratio of the total imports and exports of goods and services to the GDP | One million patacas<br>%<br>patacas<br>%<br>%<br>%<br>One million patacas<br><br>% |
| | B3—Ecological environment | C18—Water consumption per capita<br>C19—Power consumption<br>C20—Quantity of municipal solid waste disposal<br>C21—Average daily sewage treatment volume<br>C22—Total amount of waste resources recovered<br>C23—Green area per capita<br>C24—Ratio of the environmental input to the total public expenditure<br>C25—Good air rate | Liters/day<br>Million kilowatts per hour<br>Total (metric tons)<br><br>Thousand cubic meters<br><br>Metric tons (paper + plastic + metal + glass)<br>square meters<br><br>%<br><br>% |

1 MOP ≈ 0.8969 RMB (C5, C10, C12, C16). Source: The authors.

**Table 3.** Original data of the evaluation indicators of Macau's sustainable urban development from 2011 to 2018.

| Evaluation Indicators | Year | | | | | | | |
|---|---|---|---|---|---|---|---|---|
| | 2011 | 2012 | 2013 | 2014 | 2015 | 2016 | 2017 | 2018 |
| C1 | 7.3 | 9.6 | 7.9 | 8.7 | 7.9 | 7.5 | 6.8 | 5.9 |
| C2 | 18,381 | 18,993 | 19,535 | 20,518 | 21,148 | 21,393 | 20,479 | 20,777 |
| C3 | 1222 | 1354 | 1366 | 1421 | 1494 | 1591 | 1596 | 1604 |
| C4 | 2,003,949 | 1,946,457 | 2,158,707 | 1,908,109 | 2,094,188 | 2,118,728 | 2,242,195 | 2,222,880 |
| C5 | 10,000 | 11,300 | 12,000 | 13,300 | 15,000 | 15,000 | 15,000 | 16,000 |
| C6 | 416 | 417.4 | 421.3 | 424.1 | 427 | 427.4 | 427.5 | 448.9 |
| C7 | 217 | 218 | 218 | 218 | 216 | 218 | 220 | 221 |
| C8 | 2.6 | 2 | 1.8 | 1.7 | 1.8 | 1.9 | 2 | 1.8 |
| C9 | 6.9 | 1.4 | 7.3 | 2.4 | −2.6 | 5.4 | −0.7 | 0.5 |
| C10 | 293,745 | 343,416 | 413,471 | 443,298 | 368,728 | 358,200 | 404,199 | 440,316 |
| C11 | 21.7 | 9.2 | 11.2 | −1.2 | −21.5 | −0.7 | 9.9 | 5.4 |
| C12 | 534,734 | 603,495 | 697,502 | 713,514 | 564,635 | 561,858 | 627,625 | 673,481 |
| C13 | 6.4 | 6.2 | 3.7 | 5.1 | 7.8 | 6.7 | 5.1 | 4.2 |
| C14 | 93.6 | 93.8 | 96.3 | 94.8 | 92.2 | 93.4 | 94.9 | 95.8 |
| C15 | 45.4 | 45.9 | 46.1 | 58.5 | 48 | 46.7 | 49.1 | 50.5 |
| C16 | 119,263 | 151,278 | 195,770 | 218,867 | 232,447 | 250,564 | 266,729 | 292,831 |
| C17 | 58.4 | 13.6 | 23.9 | 7.2 | −26.1 | −12 | 23.9 | 15.6 |
| C18 | 351.7 | 353.4 | 353.8 | 359.5 | 359.8 | 367.3 | 371 | 373.3 |
| C19 | 3857 | 4205 | 4291 | 4469 | 4781 | 5037 | 5170 | 5319 |
| C20 | 329,992 | 365,680 | 396,738 | 457,420 | 495,331 | 502,595 | 510,702 | 522,548 |
| C21 | 186 | 203 | 215 | 217 | 193 | 230 | 211 | 223 |
| C22 | 542 | 916 | 1330 | 3989.3 | 3920.2 | 3988.4 | 3494.8 | 3608.3 |
| C23 | 15.5 | 15 | 14.5 | 14 | 13.5 | 10.9 | 10.8 | 10.6 |
| C24 | 1.8 | 1.9 | 0.9 | 1.7 | 2.1 | 2.5 | 2.5 | 2.1 |
| C25 | 0.81 | 0.69 | 0.54 | 0.45 | 0.52 | 0.46 | 0.54 | 0.64 |

Source: The authors.

If it is a positive indicator, that is, the larger the better, then use the formula $stdx_{ij}(+)$; If it is a negative indicator, that is, the smaller the better, then use the formula $stdx_{ij}(-)$. Details of the standardization of the 25 indicators are shown in Table 4.

**Table 4.** Standardized results of the indicators from 2011 to 2018.

| Evaluation Indicators | Year | | | | | | | |
|---|---|---|---|---|---|---|---|---|
| | 2011 | 2012 | 2013 | 2014 | 2015 | 2016 | 2017 | 2018 |
| C1 | 0.622 | 0.000 | 0.459 | 0.243 | 0.459 | 0.568 | 0.757 | 1.000 |
| C2 | 1.000 | 0.797 | 0.617 | 0.291 | 0.081 | 0.000 | 0.303 | 0.205 |
| C3 | 0.000 | 0.346 | 0.377 | 0.521 | 0.712 | 0.966 | 0.979 | 1.000 |
| C4 | 0.287 | 0.115 | 0.750 | 0.000 | 0.557 | 0.630 | 1.000 | 0.942 |
| C5 | 0.000 | 0.217 | 0.333 | 0.550 | 0.833 | 0.833 | 0.833 | 1.000 |
| C6 | 0.000 | 0.043 | 0.161 | 0.246 | 0.334 | 0.347 | 0.350 | 1.000 |
| C7 | 0.200 | 0.400 | 0.400 | 0.400 | 0.000 | 0.400 | 0.800 | 1.000 |
| C8 | 0.000 | 0.667 | 0.889 | 1.000 | 0.889 | 0.778 | 0.667 | 0.889 |
| C9 | 0.040 | 0.596 | 0.000 | 0.495 | 1.000 | 0.192 | 0.808 | 0.687 |
| C10 | 0.000 | 0.332 | 0.801 | 1.000 | 0.501 | 0.431 | 0.739 | 0.980 |
| C11 | 1.000 | 0.711 | 0.757 | 0.470 | 0.000 | 0.481 | 0.727 | 0.623 |
| C12 | 0.000 | 0.385 | 0.910 | 1.000 | 0.167 | 0.152 | 0.520 | 0.776 |
| C13 | 0.659 | 0.610 | 0.000 | 0.341 | 1.000 | 0.732 | 0.341 | 0.122 |
| C14 | 0.659 | 0.610 | 0.000 | 0.366 | 1.000 | 0.707 | 0.341 | 0.122 |
| C15 | 1.000 | 0.962 | 0.947 | 0.000 | 0.802 | 0.901 | 0.718 | 0.611 |
| C16 | 0.000 | 0.184 | 0.441 | 0.574 | 0.652 | 0.756 | 0.850 | 1.000 |
| C17 | 1.000 | 0.470 | 0.592 | 0.394 | 0.000 | 0.167 | 0.592 | 0.493 |
| C18 | 1.000 | 0.921 | 0.903 | 0.639 | 0.625 | 0.278 | 0.106 | 0.000 |
| C19 | 1.000 | 0.762 | 0.703 | 0.581 | 0.368 | 0.193 | 0.102 | 0.000 |
| C20 | 1.000 | 0.815 | 0.653 | 0.338 | 0.141 | 0.104 | 0.062 | 0.000 |
| C21 | 0.000 | 0.386 | 0.659 | 0.705 | 0.159 | 1.000 | 0.568 | 0.841 |
| C22 | 0.000 | 0.108 | 0.229 | 1.000 | 0.980 | 1.000 | 0.857 | 0.889 |
| C23 | 1.000 | 0.898 | 0.796 | 0.694 | 0.592 | 0.061 | 0.041 | 0.000 |
| C24 | 0.563 | 0.625 | 0.000 | 0.500 | 0.750 | 1.000 | 1.000 | 0.750 |
| C25 | 1.000 | 0.672 | 0.252 | 0.000 | 0.198 | 0.046 | 0.260 | 0.534 |

Source: The authors.

4.2.2. The Construction of the BP Neural Network

In this paper, the weight of the evaluation indicator system is determined by the BP neural network algorithm. The number of hidden layers and the number of neurons in the hidden layers are determined. The number of neurons in the input layer and output layers is selected, corresponding to the number of measurement indicators (25) and the final evaluation (1), respectively. This paper uses a single hidden layer. The number of neurons in the hidden layer can be determined by the following formula:

$$M = \sqrt{n + m} + a$$

In the above formula, N and M represent the number of neurons in the input layer and the output layer, respectively. A is the constant between [0,10]. Due to the small sample size of the study, a = 1 was taken, and the number of neurons in the hidden layer was finally determined to be six. Therefore, the structure of the BP neural network in this study is shown in Table 5.

**Table 5.** The BP neural network structure (number of neurons).

| Model Parameters | Number of Neurons in the Input Layer | Number of Neurons in the Output Layer | Number of Hidden Layers | Number of Neurons in the Hidden Layer |
|---|---|---|---|---|
| Neural Network | 25 | 1 | 1 | 7 |

Source: The authors.

### 4.2.3. Neural Network Model Fitting

Table 6 shows the path weight coefficient from the input layer neuron to the hidden layer neuron of the BP neural network. All 25 indexes of the input layer have influence on the path of seven neurons in the hidden layer, so there are 7 * 25 path weight coefficients in total.

**Table 6.** The weight coefficient from the input layer to the hidden layer.

| Indicators | The Serial Number of Neurons | | | | | | |
|---|---|---|---|---|---|---|---|
| | N1 | N2 | N3 | N4 | N5 | N6 | N7 |
| C1 | 0.040 | 0.042 | 0.018 | 0.026 | 0.002 | 0.092 | 0.065 |
| C2 | 0.093 | 0.016 | 0.092 | 0.079 | 0.058 | 0.044 | 0.026 |
| C3 | 0.075 | 0.023 | 0.006 | 0.077 | 0.067 | 0.072 | 0.064 |
| C4 | 0.042 | 0.039 | 0.082 | 0.032 | 0.081 | 0.079 | 0.085 |
| C5 | 0.051 | 0.064 | 0.095 | 0.044 | 0.006 | 0.087 | 0.063 |
| C6 | 0.036 | 0.100 | 0.022 | 0.065 | 0.060 | 0.039 | 0.014 |
| C7 | 0.003 | 0.042 | 0.018 | 0.073 | 0.037 | 0.084 | 0.073 |
| C8 | 0.057 | 0.018 | 0.096 | 0.027 | 0.092 | 0.022 | 0.037 |
| C9 | 0.009 | 0.064 | 0.018 | 0.005 | 0.072 | 0.035 | 0.066 |
| C10 | 0.038 | 0.063 | 0.002 | 0.091 | 0.080 | 0.075 | 0.081 |
| C11 | 0.038 | 0.062 | 0.058 | 0.053 | 0.028 | 0.025 | 0.045 |
| C12 | 0.023 | 0.080 | 0.099 | 0.003 | 0.054 | 0.009 | 0.080 |
| C13 | 0.099 | 0.007 | 0.094 | 0.002 | 0.068 | 0.078 | 0.053 |
| C14 | 0.089 | 0.090 | 0.063 | 0.014 | 0.022 | 0.018 | 0.004 |
| C15 | 0.011 | 0.062 | 0.094 | 0.035 | 0.041 | 0.098 | 0.095 |
| C16 | 0.068 | 0.099 | 0.077 | 0.034 | 0.066 | 0.024 | 0.030 |
| C17 | 0.068 | 0.053 | 0.041 | 0.060 | 0.075 | 0.058 | 0.055 |
| C18 | 0.058 | 0.051 | 0.008 | 0.072 | 0.100 | 0.035 | 0.097 |
| C19 | 0.035 | 0.089 | 0.045 | 0.041 | 0.022 | 0.013 | 0.031 |
| C20 | 0.073 | 0.078 | 0.069 | 0.001 | 0.084 | 0.092 | 0.077 |
| C21 | 0.004 | 0.038 | 0.070 | 0.073 | 0.022 | 0.027 | 0.067 |
| C22 | 0.048 | 0.062 | 0.024 | 0.018 | 0.083 | 0.077 | 0.093 |
| C23 | 0.011 | 0.018 | 0.010 | 0.049 | 0.019 | 0.090 | 0.010 |
| C24 | 0.004 | 0.056 | 0.077 | 0.031 | 0.018 | 0.034 | 0.021 |
| C25 | 0.051 | 0.091 | 0.063 | 0.010 | 0.039 | 0.005 | 0.050 |

Source: The authors.

Table 7 shows the path weight coefficients from seven neurons in the hidden layer to one result Indicator in the output layer, so there are 7 * 1 path weight coefficients in total.

**Table 7.** The hidden layer to the output layer weight coefficient.

| Neuron Number | N1 | N2 | N3 | N4 | N5 | N6 | N7 |
|---|---|---|---|---|---|---|---|
| weight | 0.093 | 0.092 | 0.071 | 0.062 | 0.034 | 0.094 | 0.012 |

Source: The authors.

The influence weight of the 25 indicators in the input layer on one comprehensive evaluation result in the output layer, is equal to the product of the weight matrix from the input layer to the hidden layer, and the influence weight matrix from the hidden layer to the output layer. Therefore, the weight coefficient matrix from the input layer to the hidden layer is multiplied by the weight coefficient matrix from the hidden layer to the output layer, to obtain the comprehensive weight coefficient matrix of the 25 indicators. Then, the comprehensive weight coefficient matrix is normalized to obtain the weight results of the 25 indicators and the three dimensions. The weight results of the final evaluation indicator system are shown in Table 8.

**Table 8.** Weight results of Macau's sustainable development evaluation indicator system.

| Dimension Layer | Dimension Layer Weight | Indicator Layer | Indicator Layer Weight |
| --- | --- | --- | --- |
| B1—social | 0.334 | C1—Natural growth rate of the population | 0.035 |
| | | C2—Urban population density | 0.049 |
| | | C3—Number of hospital beds | 0.042 |
| | | C4—Books | 0.046 |
| | | C5—Median monthly job income | 0.051 |
| | | C6—Total length of road lanes | 0.042 |
| | | C7—Per capita living area | 0.035 |
| | | C8—Unemployment | 0.037 |
| | | C9—Crime growth rate | 0.026 |
| B2—economic | 0.361 | C10—Gross regional product (GDP) | 0.045 |
| | | C11—GDP growth | 0.035 |
| | | C12—GDP per capita | 0.035 |
| | | C13—Second industrialization coefficient | 0.047 |
| | | C14—Third industrialization coefficient | 0.042 |
| | | C15—Gaming as a share of the GDP | 0.047 |
| | | C16—Foreign direct investment | 0.048 |
| | | C17—Ratio of the total imports and exports of goods and services to the GDP | 0.046 |
| B3—Ecological environment | 0.305 | C18—Water consumption per capita | 0.040 |
| | | C19—Power consumption | 0.034 |
| | | C20—Quantity of municipal solid waste disposal | 0.055 |
| | | C21—Average daily sewage treatment volume | 0.030 |
| | | C22—Total amount of waste resources recovered | 0.042 |
| | | C23—Green area per capita | 0.027 |
| | | C24—Ratio of the environmental input to the total public expenditure | 0.030 |
| | | C25—Good air rate | 0.036 |

Source: The authors.

*4.3. Sustainable Development of Macau*

Through the artificial neural network model fitting, each weight indicator in the sustainable development level indicator system of Macau is obtained. By multiplying the weight of these 25 indicators with their objective data, the sustainable development level score of Macau, for each year, can be obtained. The calculation formula is, as follows:

$$I = \sum_{i=1}^{25} a_i y_i$$

Among them, $I$ is the score of the sustainable development; $a_i$ is the weight value of the i-th indicator; $y_i$ is the standardized score of the i-th indicator, and the overall evaluation results are shown in the table below. It can be seen from the score that from 2011 to 2018, the score of the sustainable development in Macau increased gradually. Detailed indicators are shown in Table 9.

As seen from the change trend chart of Macau's sustainable development score below, from 2011 to 2018, Macau's sustainable development score increased steadily, then it experienced a slight decline in 2014, then recovered rapidly, and finally increased rapidly in 2017 and 2018. Detailed data are shown in Figure 4.

**Table 9.** The overall evaluation results of the index layer from 2011 to 2018.

| Measurement Indicators | Year | | | | | | | |
|---|---|---|---|---|---|---|---|---|
| | **2011** | **2012** | **2013** | **2014** | **2015** | **2016** | **2017** | **2018** |
| C1 | 0.022 | 0.000 | 0.016 | 0.009 | 0.016 | 0.020 | 0.026 | 0.035 |
| C2 | 0.049 | 0.039 | 0.030 | 0.014 | 0.004 | 0.000 | 0.015 | 0.010 |
| C3 | 0.000 | 0.015 | 0.016 | 0.022 | 0.030 | 0.041 | 0.041 | 0.042 |
| C4 | 0.013 | 0.005 | 0.035 | 0.000 | 0.026 | 0.029 | 0.046 | 0.043 |
| C5 | 0.000 | 0.011 | 0.017 | 0.028 | 0.043 | 0.043 | 0.043 | 0.051 |
| C6 | 0.000 | 0.002 | 0.007 | 0.010 | 0.014 | 0.015 | 0.015 | 0.042 |
| C7 | 0.007 | 0.014 | 0.014 | 0.014 | 0.000 | 0.014 | 0.028 | 0.035 |
| C8 | 0.000 | 0.025 | 0.033 | 0.037 | 0.033 | 0.029 | 0.025 | 0.033 |
| C9 | 0.001 | 0.015 | 0.000 | 0.013 | 0.026 | 0.005 | 0.021 | 0.018 |
| C10 | 0.000 | 0.015 | 0.036 | 0.045 | 0.023 | 0.019 | 0.033 | 0.044 |
| C11 | 0.035 | 0.025 | 0.026 | 0.016 | 0.000 | 0.017 | 0.025 | 0.022 |
| C12 | 0.000 | 0.013 | 0.032 | 0.035 | 0.006 | 0.005 | 0.018 | 0.027 |
| C13 | 0.031 | 0.029 | 0.000 | 0.016 | 0.047 | 0.034 | 0.016 | 0.006 |
| C14 | 0.028 | 0.026 | 0.000 | 0.015 | 0.042 | 0.030 | 0.014 | 0.005 |
| C15 | 0.047 | 0.045 | 0.044 | 0.000 | 0.038 | 0.042 | 0.034 | 0.029 |
| C16 | 0.000 | 0.009 | 0.021 | 0.028 | 0.031 | 0.036 | 0.041 | 0.048 |
| C17 | 0.046 | 0.022 | 0.027 | 0.018 | 0.000 | 0.008 | 0.027 | 0.023 |
| C18 | 0.040 | 0.037 | 0.036 | 0.026 | 0.025 | 0.011 | 0.004 | 0.000 |
| C19 | 0.034 | 0.026 | 0.024 | 0.020 | 0.013 | 0.007 | 0.003 | 0.000 |
| C20 | 0.055 | 0.045 | 0.036 | 0.019 | 0.008 | 0.006 | 0.003 | 0.000 |
| C21 | 0.000 | 0.012 | 0.020 | 0.021 | 0.005 | 0.030 | 0.017 | 0.025 |
| C22 | 0.000 | 0.005 | 0.010 | 0.042 | 0.041 | 0.042 | 0.036 | 0.037 |
| C23 | 0.027 | 0.024 | 0.021 | 0.019 | 0.016 | 0.002 | 0.001 | 0.000 |
| C24 | 0.017 | 0.019 | 0.000 | 0.015 | 0.023 | 0.030 | 0.030 | 0.023 |
| C25 | 0.036 | 0.024 | 0.009 | 0.000 | 0.007 | 0.002 | 0.009 | 0.019 |
| score | 0.487 | 0.500 | 0.510 | 0.481 | 0.514 | 0.515 | 0.573 | 0.617 |

Source: The authors.

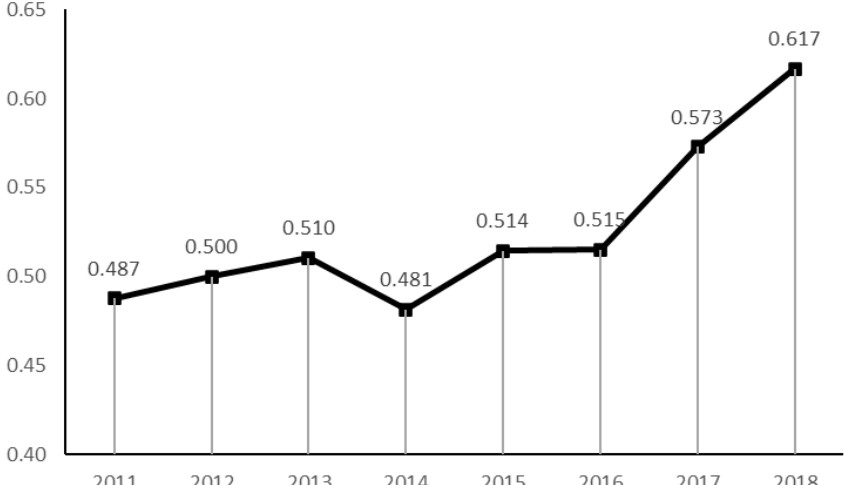

**Figure 4.** Line chart of Macau's sustainable development score from 2011 to 2018 (ANN result). Source: The authors.

*4.4. Correlation Analysis of Macau's Economic and Ecological Environment Development*

According to the result data of the BP neural network model (Table 10), the Pearson correlation coefficient of the economic dimension and the ecological environment dimension is r = −0.832, $p = 0.010 < 0.05$. (Among them, R stands for the correlation coefficient, which ranges from −1 to 1. When r = 1, it means that they are completely correlated; when r = −1, it means that they are negatively correlated; when R = 0, it means that they

are entirely unrelated. The $p$ value represents the significance of the correlation. When the $p$ value is less than 0.05, the correlation between the two is significant.) The results of r and $p$ show a significant negative correlation between Macau's economy and ecological environment.

**Table 10.** Correlation analysis of the economic and ecological environment indicators in Macau.

| Dimension 1 | Dimension 2 | r | p |
|---|---|---|---|
| Economic | Ecological environment | −0.832 | 0.010 |

Source: The authors.

According to the overall evaluation results of the indicator layer, the ecological indicators and economic indicators of each year were added together to obtain the ecological and economic scores for each year from 2011 to 2018. The broken line chart of each year was drawn, based on the scores (Figure 5).

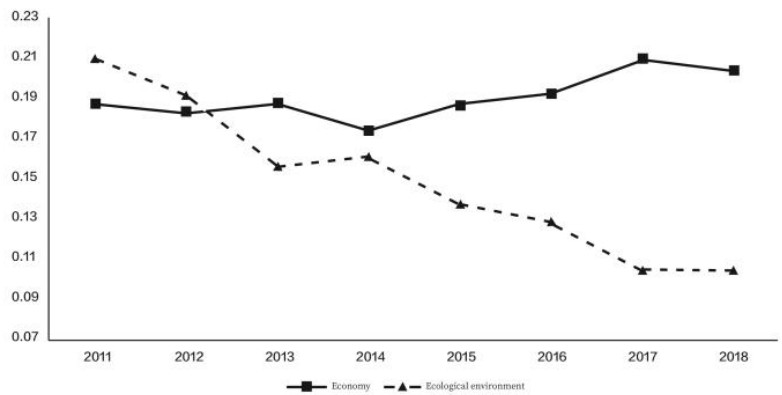

**Figure 5.** Economic and ecological environment score year-line chart. Source: The authors.

According to the year line chart of Macau's economic and ecological environment scores, we can see the relationship between Macau's ecological environment and economy. The line chart shows that in years with better economic scores, the eco-environment scores were lower.

## 5. Discussion

According to the order of weight, the order of social, economic, and ecological environment is that: the economy takes the advantage, society comes in second place, and the ecological environment is the least prioritized. This result shows that Macau attaches importance to the economic development in the sustainable urban development, while the research on the development of the social and ecological environment is relatively weak. The development of the ecological environment is the factor that Macau needs to pay more attention to, in future urban development.

According to the comprehensive weight of each indicator, Macau's economy is developed. This is evident in the gambling industry, which has brought substantial profits to Macau's economy, and it is a typical economically developed city.

The coordinated development among the subsystems, especially the economic, social and environmental subsystems, is an important part of the urban sustainability and directly affects the quality of urban development [81]. Macau should also follow the principle of coordinated development. The economic development of Macau directly affects the score of the sustainable urban development of Macau, but there are problems, such as a high population density and limited green space. Macau can be positioned as a high-density developed city, similar to European countries with a good sustainable development. The difference, however, is in the consumption of resources. Compared with developed countries, the development cost of Macau is much higher than these countries. To realize the economic

value, the environmental pollution of the development process is higher than that in European and American countries, such as the United States and Britain. For example, China's pollution emissions under the condition of the per capita GDP, USD 400~1000, is equivalent to the pollution produced by an average of USD 3000~10,000 in European and American countries, and they have made sustainable development the goal [82]. Given the above, the planning of European countries, based on the concept of sustainable development, is worth learning from. In the 1990s, some European countries, such as the United Kingdom, Finland, and Ireland, successively formulated their own sustainable development strategies. In 1997, sustainable development was formally written into the Amsterdam Treaty, as one of the primary objectives of EU policy [83]. Many European countries were early to recognize the profound relationship between environmental protection and economic and social development. For example, the UK has formulated sustainable development strategies and action plans at the national, regional, local, and community levels, and has implemented sustainable development strategies as the behavior of the whole society [84]. Finland has been committed to a strategy of sustainable development driven by a circular economy. The circular economy, which emphasizes sustainability through the reduction, reuse, and recycling of products and materials from production, distribution, and consumption, is also one of the ways to alleviate poverty. [85]. Germany has made environmental protection and sustainable development a national strategy. The German Sustainable Development Strategy was adopted on 10 March 2021, with 17 Sustainable Development Goals and the principles of action "people, planet, prosperity, peace and partnership" as the government's working principles. [86].

From the perspective of Macau's ecological and economic relationship, there is a significant negative correlation between Macau's economy and ecological environment. This is also reflected in the economic and ecological environment scores, especially in the years with better economic scores, the ecological environment scores are significantly lower. The main reason lies in the further compression of the ecological environment space caused by large-scale urban engineering construction. For example, the rise of a new round of reclamation projects in Macau has promoted the development of the city's economy, but has brought damage to the ecological environment. The economic development pattern of Macau can also account for this reason. With the gradual prosperity of Macau's tourism industry, the Macau government attaches more importance to the development mode focusing on tourism, such as the gambling industry and the historical city with Chinese and Portuguese cultural customs [87,88]. In the draft of the master plan for *the City of the Macau Special Administrative Region (2020–2040)*, Macau is positioned as *with Chinese culture as the mainstream, and a multi-cultural exchange and cooperation base* and *the central city of the Guangdong-Hong Kong-Macau Greater Bay Area* [89]. However, in the process of economic development, it neglects the "mountain and sea" of Macau, which is the unique natural environment of the city [90], so the economic and ecological environment is polarized, year by year.

Macau's ecological environment and economic development do not present a desirable and synergistic relationship, which is not a sustainable development state for the urban development. The reason for the mismatch, is that Macau's economy has grown so fast in the past decade, Macau's per capita income has surpassed that of Hong Kong, making it one of the richest regions in the world [63]. Economic development is accompanied by the high-speed operation of economic activities. In the economic wealth accumulation, Macau constantly claims ecological and environmental resources in various forms, resulting in the rapid consumption of natural resources over time. Household waste and sewage treatment have become problems to be solved on the road to sustainable urban development [90]. As John Ormsbee Simonds points out in the Earthscape, the concept of co-aesthetic should be established to push the ecological landscape research to "studying the human living space and vision" [91]. Macau can draw lessons from this concept.

## 6. Conclusions

This paper selects the BP neural network to establish a set of sustainable urban development evaluation index systems from three levels of the urban economy, ecology, and society, and makes an objective analysis of Macau's sustainable development situation. Macau's sustainable urban development should not only focus on the economic development, but also pay attention to the coordinated development of the ecological environment. The ecology, economy, and society should be treated as an organic whole and aim to achieve a sustainable urban development through government regulation, scientific and technological breakthroughs, the popularizing of sustainable development ideas, and other methods. Meanwhile, from the perspective of synergy, the degree of coordination between ecology and economy in Macau should be consistent, so as to better meet the needs of sustainable development. In recent years, with the continuous improvement of the scientific and technological levels of the gambling industry, and the continuous optimization of the tourism industry structure, Macau is moving towards the goal of becoming a world leisure and tourism center. Although gambling tourism still occupies a large proportion in the economic and industrial structure, Macao gradually diversifies its economy. The experience can be learned from Las Vegas, Monte Carlo, and other cities that also rely on gambling, as an important pillar industry. Their economy has developed into a diversified model integrating gambling, exhibition, leisure, shopping, and other industries. The successful transformation of their economies, from unitary to diversified, provides valuable experience for Macau. Macau is in urgent need of using gambling tourism to attract tourists and promote the development of various service industries, such as exhibition, catering, shopping, transportation, and sightseeing in the surrounding historic city. In general, while developing its economy, Macau should improve its ecological environment construction as soon as possible. Macau's construction should focus on improving the urban environment, enhance the comprehensive strength of the city, and improve the visual beauty of the city, to achieve a match between the ecological, economic, and society development, to finally realize the real sense of sustainable development.

**Author Contributions:** Conceptualization, Y.H. and S.Y.; methodology, Y.H.; software, Y.T.; validation, Y.H. and S.Y.; formal analysis, Y.H.; investigation, S.Y.; resources, Y.H.; data curation, S.Y.; writing—original draft preparation, Y.H.; writing—review and editing, S.Y.; visualization, S.Y.; supervision, Y.H.; project administration, Y.H.; funding acquisition, S.Y. All authors have read and agreed to the published version of the manuscript.

**Funding:** Funded by the China Postdoctoral Science Foundation (Grant: 2021M692781).

**Institutional Review Board Statement:** Not applicable.

**Informed Consent Statement:** Not applicable.

**Data Availability Statement:** The data presented in this study are available in this published article.

**Acknowledgments:** The authors acknowledge the help from the university colleagues. Compliance with Ethical Standards.

**Conflicts of Interest:** The authors declare no conflict of interest.

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
