# Peer review of "Evaluation of the Sustainable Development of Macau, Based on the BP Neural Network"

_sustainability, doi:10.3390/su15010879_

Round 1

Reviewer 1 Report

I would like to see this work in publication. However, it has some clear issues that need to be addressed first.  

  1. The methodology is interesting and provides useful information. There is sufficient information to explain the method. However, there is insufficient justification for using the BP neural network model as opposed to some other model. There should be a justification explaining the advantages of the approach over alternatives.  
  2. The literature review seems heavy on deeper urban background, with historical references like Mumford and reference to Le Corbusier, but significantly less mention of more recent sustainability studies, except to sort of list some general ideas. The important departure of adapting synergetics to this area requires a deeper explanation of the utility of synergetic explanations of sustainable urban development above other approaches.  
  3. The application of synergetics is not entirely clear. At times, the authors appear simply to mean “synergy” and at others they suggest the field of study represents a theory. The meaning is unclear in this context and I suspect that a clearer explanation of this as an approach (or theory if they explain how it is used as theory) would go a long way to repairing the literature and clarifying the purpose of using a BP neural network model.  
  4. There are frequent English errors, some requiring a close reading for the correct understanding of the passage, in addition to numerous errors and typos.  
  5. There are also a few more minor concerns: 
    1. (1:25) The reference to the industrial revolution “Around the 18th Century...” is somewhat unclear followed by the Mumford quote emphasizing damage done in the 19th Century. Perhaps something specifying later in the 18th Century? 
    2. (1:34-35) The MAB is the “Man and Biosphere Programme.” 
    3. (3:108-109) Haken’s field of study is not “synergetic” (and adjective), but “synergetics.” 
    4. (3:120) What are the “materials” provided by Macau beyond the data? 
    5. (6:225) Table 1. has some issues: “urban construction” should be “urban infrastructure;” I am not sure what is meant by “to both GDP;” “Second industrialization” should be “Secondary industry;” “Third industrialization” should be “Tertiary industry;” It should be “Total imports and exports of goods and services as a share of GDP” and could be shortened to “Import/export as a share of GDP.” 
    6. (7:237) Likewise with Table2.: C3 “zhang” should be explained in terms of square meters for those without an understanding of Chinese measurements. C4 “Copies of the...” should simply be “number” or “volumes.” C7 “Ft.” is unclear. C5, C10, C12, C16 should have a footnote explaining pataca and the relationship to major currencies as a reference. C23 should be expressed as square meters. C25 “Macau north” needs some explanation.  
    7.  

Author Response

请看附件,谢谢。

Reviewer 2 Report

The authors do not seem to have put much effort in the editing of the manuscript nor its coherence. As a result, it flouts some of the most basic rules in academic writing. The manuscript is also missing some crucial contextual information that will show the readers why the study needed to be conducted and against which background it was conducted.

The authors also use relatively old sources to reference their work. At least 50% of referenced literature needs to have been published in the last 10 years (2012-2022). Currently, only 7 of the consulted literature were published in this period.

Sustainability is an international journal. That means the research they publish should not only reach an international audience, but also be relatable. Currently, this manuscript is only focused on Macau and to an extent, China. As it is, how does it relate to research in South America, Africa, India, etc? The Discussion does not make much reference to international literature.

Currently, I do not see its contribution to knowledge in the broader SD literature. There are also no conclusions to suggest that the authors may have attempted to make such contributions, in case a reader might have missed them.

Should the authors opt to resubmit, they should read the GUIDE TO AUTHORS thoroughly, and make sure that the manuscript follows those guidelines.

Author Response

Please see the attachment,thank you.

Reviewer 3 Report

The author's work is worthy of recognition, but there are still some problems to be rectified.

1. The article lacks the literature review chapter, so it is suggested to supplement it. 2. The theoretical part of the research needs to be divided into a separate section, and it should be described emphatically. 3. It is suggested that the author use professional drawing software to draw pictures instead of simply pasting them from other places. 4. It is suggested to increase theoretical support for index selection. Please refer to and quote this article :Exploring Regional Advanced Manufacturing and Its Driving Factors_ A Case Study of the Guangdong–Hong Kong–Macao Greater Bay Area 5. The calculation process of bp neural network is a "black box". How does the author verify that the training process and output results of BP neural network are effective? In the field of prediction and classification, error indicators are usually selected for verification. But obviously, this article is not applicable, which requires the author to think carefully and give a reasonable explanation. 6.discussion and conclusion should be separated.

Author Response

Please see the attachment,thank you.

Round 2

Reviewer 1 Report

I am satisfied with the paper in its current form. I would still suggest a more careful proofreading.

Author Response

Dear reviewer,
Thank you for your support of my manuscript. In order to make this paper more logical and perfect, we have not only corrected the grammatical errors, but also added the research scope and the content of sustainable development of Macau in the literature review. 

Thank you for your comments again.

Reviewer 2 Report

In addition to the attached document, authors should take note of the following:

1) The phrase "put forward" is overly and unnecessarily used. Authors need to find alternative phrases.

2) The phrase "scholars" or "academics" are also overly used. Refer to scholarly research on... instead.

3) The literature review lack "meat". In its current form is is simply a reiteration of what some institutes and organizations have said about sustainable development. There is nothing that links to this current study. The literature review needs a lot of improvement.

4) After the literature review, so what? What is the point of all that information? Where are the research questions or hypotheses? Where is the description of the study area?

Reviewer 3 Report

The author has made changes and his work is worthy of recognition.

Author Response

(The authors gave the same response as above.)
